# Post-Translational Modifications of the Energy Guardian AMP-Activated Protein Kinase

**DOI:** 10.3390/ijms22031229

**Published:** 2021-01-27

**Authors:** Ashley J. Ovens, John W. Scott, Christopher G. Langendorf, Bruce E. Kemp, Jonathan S. Oakhill, William J. Smiles

**Affiliations:** 1Metabolic Signalling Laboratory, St Vincent’s Institute of Medical Research, School of Medicine, University of Melbourne, Fitzroy, VIC 3065, Australia; aovens@svi.edu.au (A.J.O.); joakhill@svi.edu.au (J.S.O.); 2Mary MacKillop Institute for Health Research, Australian Catholic University, Fitzroy, VIC 3000, Australia; jscott@svi.edu.au (J.W.S.); bkemp@svi.edu.au (B.E.K.); 3Protein Chemistry & Metabolism, St Vincent’s Institute of Medical Research, School of Medicine, University of Melbourne, Fitzroy, VIC 3065, Australia; clangendorf@svi.edu.au; 4The Florey Institute of Neuroscience and Mental Health, Parkville, VIC 3052, Australia

**Keywords:** AMPK, energy metabolism, phosphorylation, ubiquitination, oxidation

## Abstract

Physical exercise elicits physiological metabolic perturbations such as energetic and oxidative stress; however, a diverse range of cellular processes are stimulated in response to combat these challenges and maintain cellular energy homeostasis. AMP-activated protein kinase (AMPK) is a highly conserved enzyme that acts as a metabolic fuel sensor and is central to this adaptive response to exercise. The complexity of AMPK’s role in modulating a range of cellular signalling cascades is well documented, yet aside from its well-characterised regulation by activation loop phosphorylation, AMPK is further subject to a multitude of additional regulatory stimuli. Therefore, in this review we comprehensively outline current knowledge around the post-translational modifications of AMPK, including novel phosphorylation sites, as well as underappreciated roles for ubiquitination, sumoylation, acetylation, methylation and oxidation. We provide insight into the physiological ramifications of these AMPK modifications, which not only affect its activity, but also subcellular localisation, nutrient interactions and protein stability. Lastly, we highlight the current knowledge gaps in this area of AMPK research and provide perspectives on how the field can apply greater rigour to the characterisation of novel AMPK regulatory modifications.

## 1. Introduction

Adenosine triphosphate (ATP) is the energy currency of the cell containing energy-rich phosphoanhydride bonds that fuel a broad range of biochemical processes such as cellular signalling, protein synthesis, cell division and muscle contraction. Therefore, its concentration is tightly controlled such as in skeletal muscle where at baseline ATP levels are approximately 8 mM [1,2]. Despite physical exercise causing upwards of a 1000-fold increase in ATP demand [3], its concentration remains relatively constant without falling below approximately 6 mM [4]. Tight regulation of ATP levels is indeed crucial for the maintenance of cellular metabolism and cell survival, as prolonged disturbances to cellular energy pools inevitably leads to programmed cell death [5], a process underpinning several metabolic disease states (e.g., cardiovascular disease, type II diabetes etc.). However, acute metabolic perturbations elicited by strenuous physical exercise that is reinforced by repeated training bouts, underpins the therapeutic benefits of this physiological stimulus toward human health. From a molecular standpoint, these health benefits are underscored by the activation of a diverse range of signal transduction cascades whose *modus operandi* is essentially to restore homeostasis in response to metabolic stress. The net result of repeated exercise-induced fluctuations in these pathways is an increase in the steady-state abundance of requisite signalling proteins (i.e., enzymatic) and energy availability (i.e., sugars, triglycerides) to better combat future metabolic disturbances [6]. This phenomenon is perhaps best exemplified by a process termed glycogen supercompensation. Here, a single exhaustive exercise stimulus depletes muscle glycogen stores, whereby with adequate recovery and dietary carbohydrate consumption, glycogen levels are replenished to a greater extent than that of the previous baseline (thus adaptation) in a manner governed by several metabolic enzymes [7,8].

Central to the propagation of these exercise-induced signalling events is the activity of AMP-activated protein kinase (AMPK), the energy guardian or metabolic “fuel gauge” of the cell [9,10,11,12]. As a whole, AMPK upregulates catabolic, nutrient/ATP-generating processes such as fat oxidation, glucose uptake and autophagy, while restraining anabolic, ATP-consuming processes such as protein and lipid synthesis. AMPK is a phylogenetically conserved heterotrimeric serine/threonine protein kinase comprised of α-, β- and γ-subunits. The α-subunit houses the catalytic kinase domain, and a COOH-terminal region that contains regulatory and scaffolding domains (Figure 1). The β-subunit also contains regulatory and scaffolding domains, while the γ-subunit possesses an NH_2_-terminal scaffolding domain followed by four cystathionine β synthase (CBS) repeats that arrange into two Bateman domains (Figure 1) [13]. Ultimately, these form four adenine nucleotide binding sites (sites 1–4) that give AMPK its energy sensing capabilities: sites 1 and 3 can exchangeably bind AMP, ADP and ATP; site 2 lacks a critical Asp residue rendering it incapable of binding nucleotides; site 4 is thought to permanently house AMP. Several isoforms exist for each subunit (α1/2, β1/2, γ1/2/3), giving rise to 12 possible configurations expressed differentially in various cells and tissues [9,11,14]. Interestingly, AMPK complexes expressed in skeletal muscle are selectively activated in response to different exercise regimes. For example, the α2β2γ3 complex that is exclusively expressed in skeletal muscle (predominantly in white, glycolytic (fast twitch/type II) fibres [15]) is preferentially activated by high-intensity exercise [16,17]. While α1β2γ1 and α2β2γ1 complexes, which are not limited to expression in skeletal muscle, are stimulated by more prolonged, endurance-based forms of exercise in red, oxidative (slow twitch/type I) skeletal muscle fibres [18]. This isoform-specific coordination of AMPK activation in response to exercise is intriguing, and although no studies have investigated the cellular mechanism, it is tempting to speculate that this may be achieved through a diverse range of exercise-induced post translational modifications (PTMs) on the specific AMPK heterotrimers.

In a physiological context, AMPK has a well-known ability to stimulate cellular glucose uptake. Although AMPK was originally thought to be responsible for augmenting insulin-independent glucose uptake into skeletal muscle during exercise, there is mounting evidence suggesting this is not the case and it instead plays a more pivotal role in post-exercise glucose uptake and subsequent glycogen resynthesis [16,19,20]. This specific context for AMPK’s role in promoting glucose uptake also points to a novel set of exercise stimulated PTMs controlling the time and place of AMPK activation. In addition to the benefit of AMPK inducing glucose disposal, when mice are placed on a high fat diet the muscle-specific γ3-subunit is known to be vital for preventing the accumulation of triglycerides and protecting against insulin resistance [16]. It is therefore unsurprising that AMPK, in particular skeletal muscle-enriched complexes such as α2β2γ3, have emerged as attractive therapeutic targets for the treatment of metabolic disease. Unfortunately, recent efforts at pharmacologically targeting AMPK directly using small-molecule pan activators (i.e., those activating all 12 AMPK complexes) have been confounded by significant side-effects such as cardiac hypertrophy, precluding any clinical viability [21]. This is presumably, in part, a consequence of activating γ2-containing AMPK complexes that are highly expressed in the heart, for which point mutations in human γ2 triggers constitutive activation and a disease phenotype manifesting in left ventricular hypertrophy and excess glycogen storage [22,23]. Conversely, loss of γ2 in the heart increases resting heart rate and abrogates the cardiac benefits associated with endurance exercise training [24]. These complex physiological consequences of AMPK activity once again highlight the importance of the time and place of AMPK activation. Despite the deleterious long-term effects of pan AMPK activation, the efficacy of these compounds in improving the hallmarks of metabolic disease is becoming well documented [21,25]. Hence, future efforts in this area should be focussed on developing isoform-specific AMPK activators in an effort to avoid systemic AMPK activation and promote activation in tissues with high metabolic activity such as liver and skeletal muscle. One way of doing this would be to specifically target β2-containing complexes that are primarily expressed in these tissues [26,27,28], or γ3-containing complexes that, as already mentioned, are exclusively expressed in skeletal muscle [15].

At the molecular level, AMPK is subject to a plethora of dynamic PTM events as expected of a major nutrient-sensing hub [29]. Functional roles for a handful of PTMs, such as activation loop phosphorylation, are well documented and represent pioneering work in the field. When set against the >150 isoform-wide phosphorylatable residues on AMPK, however, the inadequacy of our knowledge depth in certain areas becomes apparent. As such, knowledge of how each AMPK complex is regulated is not only crucial for effectively designing isoform-specific drugs to treat metabolic disorders, but it will better inform us in which physiological context AMPK activation will be most advantageous. Here, we focus on the current knowledge of reversible AMPK PTMs and their implications for future investigations.

## 2. Regulation by Phosphorylation

Phosphorylation events on AMPK with known function are summarised in Table 1 with their structural position highlighted in Figure 1. Additional sites that have been identified in high-throughput (HTP) or in vitro studies (cell-free using purified enzymes or in cellulo), but are yet to undergo official validation or functional analysis, are summarised in Table 2.

### 2.1. α-Subunit

#### 2.1.1. Activation Loop

The α-subunit kinase domain possesses an archetypal kinase domain structure consisting of two lobes, a small N-lobe and larger C-lobe. The C-lobe contains a series of essential residues collectively termed the activation loop (α1: 159–185; α2: 157–183). Phosphorylation of protein kinase activation loops is critically important for their activity, as the negatively charged phosphate group coordinates both the active site and substrate binding groove into a catalytically-favourable conformation [63]. In AMPK, this loop includes the conserved threonine residues **α1-T174**/**α2-T172** (henceforth α-T172), which align with the activation loop sites on numerous other typical eukaryotic protein kinases. α-T172 was discovered to be phosphorylated over 20 years ago and was shown to lead to >50-fold activation of AMPK [30]. Phosphorylation of α-T172 is enhanced upon binding of AMP or ADP to γ-subunit sites 1 and 3 provided the β-subunit is NH_2_-terminally myristoylated, while AMP and ADP binding can also protect α-T172 from phosphatase dephosphorylation independent of myristoylation status [64,65,66]. The identity of the α-T172 upstream kinase remained enigmatic for many years until it was revealed in 2003 that the liver kinase B1 complex (LKB1/STRADα/β/MO25α/β) performed this role *in vitro*, and was required for the phenformin- (a mitochondrial complex I inhibitor) and 5-aminoimadazole-4-carboxamide-1-β-D-ribofuranoside- (AICAR, a membrane-permeable version of the AMP mimetic ZMP) induced phosphorylation of α-T172 in multiple mammalian cell lines [31,32,67]. Interestingly, the lysosome, which not only is firmly established as a major hub for mammalian (mechanistic) target of rapamycin complex 1 (mTORC1) activation [68,69], has also emerged in recent years as a site of AMPK regulation depending on the prevailing nutrient and hormonal milieu [70]. Current models suggest that AMPK and mTORC1 share identical resident lysosomal scaffolds, specifically the v-ATPase/Ragulator complex, yet with distinct binding partners and LKB1 as the lysosomal α-T172 kinase [70,71]. The latter is significant, since LKB1 is the preferred α-T172 kinase activating skeletal muscle α2-containing AMPK in response to high-intensity exercise [18]. However, LKB1-deficient cells retain residual phosphorylation on α-T172, suggesting an alternative upstream kinase(s). Indeed, calcium/calmodulin-dependent kinase kinases (CaMKKs) were shown to fulfil this role, whereby both CaMKK1 and CaMKK2 activated full-length heterotrimeric α1β1γ1 [72] and GST-α1 [33] in cell-free assays. Notably, in a variety of mammalian cell lines, exposure to Ca^2+^ ionophores stimulated α-T172 phosphorylation in an LKB1-null background, an effect abolished by the CaMKK inhibitor STO-609 [33,34,72]. siRNA knockdown of CaMKK2, but not CaMKK1, reduced basal AMPK activity in LKB1-null HeLa cells, suggesting CaMKK2 is the physiologically relevant α-T172 kinase in vivo [33,35,36].

Other kinases have been proposed to activate full-length AMPK in vitro by α-T172 phosphorylation, such as transforming growth factor-β-activated kinase (TAK1) [35,36]. Overexpression of TAK1 in HeLa cells increased α-T172 phosphorylation, with a similar response following exposure to TAK1-activating cytokines such as tumour necrosis factor-α (TNFα) and interleukin-1. Although the physiological relevance of TAK1/AMPK signalling has been questioned [73], recent evidence suggests that in certain cellular contexts (e.g., during lysosomal damage or cell infection) TAK1 phosphorylation of AMPK α-T172 is indeed meaningful [73,74,75]. More recently, two additional α-T172 kinases were proposed, including mixed-linkage kinase 3 (MLK3) [37] and vaccinia virus-related kinase 1 (VRK1) [38]. However, both of these studies only verified this regulation via the use of recombinant α-subunit and not the full AMPK heterotrimer. Hence, the weight of evidence in favour of MLK3 and VRK1 as α-T172 upstream kinases remains equivocal with further verification warranted, particularly in a physiological context. Compelling evidence indicates that **α1-S175/α2-S173**, adjacent to α-T172, can be phosphorylated by protein kinase A (PKA) to inhibit α-T172 phosphorylation and reduce AMPK activity [39]. This mechanism was investigated by cell-free assay using full-length α1β1γ1, and in adipocytes. PKA-mediated phosphorylation of α-S173 relieved AMPK’s inhibition of hormone sensitive lipase (HSL), thereby enabling PKA to further activate HSL and promote lipolysis [39]. Two subsequent studies have provided further physiological context for α-S173 phosphorylation in promoting cell death [76] and insulin resistance [77], and this modification has been detected in multiple HTP studies [29].

Inactivation of AMPK is an equally important and tightly controlled aspect of its regulation. This mechanism involves the dephosphorylation of α-T172 by the protein phosphatases; protein phosphatase 2A (PP2A), protein phosphatase 2C (PP2C) and Mg^2+^/Mn^2+^-dependent protein phosphatase 1E (PPM1E) [64,78]. Protein phosphatase regulation not only inactivates AMPK but can also modulate the extent of AMPK activation by working in concert with upstream α-T172 kinases LKB1 and CaMKK2 [9]. Whether α-T172 phosphorylation is absolutely required for AMPK activity is disputed. Allosteric drugs, such as A769662, can synergise with AMP, or the AMP mimetic compound C2, to stimulate AMPK activity >1000 fold independently of α-T172. However, the physiological relevance of signalling from this appreciable AMPK cellular pool requires further study [56,79,80]. Furthermore, many studies use α-T172 as a marker for AMPK activity, which can be misleading. For example, the dual AMPK and autophagy initiator ULK1 inhibitor SBI-0206965 counterintuitively induces an increase in α-T172 phosphorylation, possibly due to de-repression of ULK1 negative feedback through this site [61], while suppressing AMPK signalling [81,82]. This highlights the importance of using downstream AMPK substrates as markers of AMPK activity instead of α-T172 phosphorylation.

#### 2.1.2. α-Linker

Immediately following the α-subunit kinase domain lies a tri-helical autoinhibitory domain (AID; α1: 290–337; α2: 288–335) [83,84] and the α-linker (α1: 334–395; α2: 332–399) which contains two tandem α-regulatory subunit interacting motifs (α-RIMs) [85]. The α-RIM1 sequence (α1: 334–344; α2: 332–342) interacts with the nucleotide-free γ-subunit site-2, while α-RIM2 (α1 residues 360–370; α2 residues 364–374) forms hydrogen bonds with the AMP-bound γ-subunit site-3, thereby directly sensing the AMPK-bound nucleotides [47]. Interestingly, immediately following α-RIM1 lies **α1-S347/α2-S345** (henceforth α-S345), which is phosphorylated in C2C12 myotubes, mouse liver and human vastus lateralis muscle [40,41], and has been identified in multiple HTP phosphoproteomic studies [86,87,88]. Cyclin-dependent kinase 4 (CDK4) was shown to phosphorylate α-S345 in a cell-free assay using fragments of the α2 subunit as substrate, although this effect could not be reproduced using full-length heterotrimeric AMPK [40]. We recently demonstrated that mTORC1 is an upstream kinase for α-S345 in vivo [41]. Specifically, mTORC1 phosphorylation on α-S345 downregulated α-T172 phosphorylation in several mammalian cell lines (e.g., C2C12, HEK293T, mouse embryonic fibroblasts (MEFs)), inhibiting AMPK to promote cell proliferation. Importantly, we uncovered an ancient (conserved from fission yeast to humans) negative feedback loop between mTORC1 and AMPK; for several decades AMPK has been known to either directly or indirectly inhibit mTORC1 by phosphorylation of Raptor [89] and TSC2 [90], respectively, constituting a metabolic checkpoint in the face of limited nutrient supply. In human exercise studies immediately following intense exercise contraction there is a blunting of anabolic muscle protein synthesis commensurate with elevated α2-complexed AMPK activity [91]. As AMPK activity begins to plateau, protein synthesis rates peak, whereby this reduction in α2-complexed AMPK activity in particular parallels elevated mTORC1 signalling [92], an event rate-limiting for contraction-mediated protein synthesis in humans [93] and to a lesser extent in rodents [94]. However, the mechanism by which this feedback loop controls protein synthesis is unclear. Notwithstanding said differences between humans and rodents, pronounced mTORC1 signalling and resultant protein synthesis is detected well into recovery (4 h) following resistance-like exercise in mice despite persistent AMPK phosphorylation of Raptor [95], a substrate targeted by AMPK even in the face of mild energetic stress [96], suggesting this branch of AMPK signalling does not impinge upon exercise-induced anabolic adaptations. In fact, mice in that study [95] expressing kinase dead α2-containing AMPK have a blunted muscle protein synthetic response to the exercise stimulus that is rescued by normalising an intramuscular glycogen defect. It is worth noting that Yoon and colleagues [97] recently found that glucose restriction and repression of mTORC1 signalling (albeit in HEK293T cells) occurs specifically via AMPK activation of ULK1 and inhibition of leucyl-tRNA synthetase 1, an enzyme that covalently couples leucine to its cognate tRNA using ATP, signifying an energy conservation process involving upregulation of autophagy, a principal cellular energy harvesting pathway. Altogether this is potentially reminiscent of the metabolic scenario encountered immediately following exercise [91]. An α-subunit isoform-specific role in this process appears evident since unlike α2, AMPK α1-subunit knockdown accelerates muscle growth incurred by mechanical overload in mice [98]. Since protein synthesis is one of the most energetically demanding processes encountered by the cell (4 high-energy phosphates required per peptide bond during translation), α2-complexed AMPK in particular likely functions to rapidly replenish cellular energy reservoirs via upregulation of energy-generating (i.e., autophagy, glucose uptake) processes in the immediate period following exercise [19] to in turn, facilitate the transition to mTORC1-driven, ATP-consuming adaptive events (i.e., protein synthesis). Whether mTORC1 phosphorylation of α2-S345 alone, or in combination with other undiscovered PTMs on human skeletal muscle AMPK, is implicated in the fine-tuning of this intricate process is a point of future study.

#### 2.1.3. β-SID and ST Loop

At the extreme COOH-terminus of the α-subunit lies a β-subunit interacting domain (β-SID; α1: 396–550; α2: 400–552) that is crucial for stabilising the entire AMPK complex [99]. The majority of the β-SID is conserved between α1/α2 isoforms and folds into an ordered structure comprising two α-helices and an anti-parallel β-sheet containing four β-strands, however it also contains a highly divergent and unstructured ~60 amino acid serine/threonine-rich loop (ST loop; α1: 471–530; α2: 475–532) which is highly regulated by a number of phosphorylation sites. Situated just prior to the ST loop and located on the first turn of the β-sheet is a solvent exposed **α1-Y432/α2-Y436** (hereafter α-Y436) that was found to be phosphorylated in skeletal muscle by Fyn, a member of the Src family of protooncogene non-receptor tyrosine kinases with key roles in inflammation [42]. TNFα cytokine stimulation of Fyn attenuated AMPK-mediated autophagy due to direct α-Y436 phosphorylation that blunted AMPK activity [42], an effect that is independent of LKB1. Cell-free phosphorylation of α-Y436 by Fyn was confirmed using an isolated α2-subunit and a generic phospho-tyrosine antibody, hence further validation with heterotrimeric AMPK is required.

At the start of the ST loop is **α1-T481/α2-T485** (hereafter α-T481), which was shown to be phosphorylated in cellulo and in cell-free assays by glycogen synthase kinase 3 (GSK3)α and GSK3β [43], while α2-T485 is also phosphorylated by CDK4 in cell-free assays [40]. Phosphorylation here imposed an inhibitory effect on AMPK by promoting α-T172 dephosphorylation. Specifically, the authors proposed that a dephosphorylated ST loop physically obstructs the α-subunit kinase domain, in turn protecting α-T172 from phosphatases, whereas phosphorylation causes the ST loop to dissociate from the kinase domain, providing phosphatase accessibility to the region. This hypothesis was supported by evidence showing a polypeptide ST loop fragment co-immunoprecipitated with the α-subunit kinase domain harbouring the non-phosphorylatable α-T481A mutant, whereas this was abolished with the phosphomimetic α-T481E. In addition to α-T481, GSK3 was shown to phosphorylate **α1-S477** and **α1-T473** to create a phosphorylation cluster dependent on prior “priming” phosphorylation on α-T481, characteristic of many GSK3 substrates [100]. However, the physiological relevance of these additional phosphorylation events remains uncertain, as substitution to Ala residues had no bearing on GSK3′s ability to reduce α-T172 phosphorylation. Interestingly, phosphorylation of α-T481 by GSK3 was shown to be dependent on prior phosphorylation on **α1-S487/α2-T491** (hereafter α-S487), however this was not a typical GSK3 priming event, as the GSK3β R96A mutant which blocks GSK3 phosphorylation of primed but not non-primed substrates, is still capable of inducing α-T481 phosphorylation [43,101]. Instead, the authors suggested that the reliance on α-S487 phosphorylation may be due to a conformational change that exposes α-T481 and allows GSK3 access to the site.

Phosphorylation of α-S487 was first reported in 2003 on bacterially expressed α1β1γ1, however this site was originally labelled α1-S485/α2-S491 as the α1 sequence used was a noncanonical rat sequence missing α1-T3 and α1-A4 [44]. α1β1γ1 treated with a partially purified rat liver AMPK kinase preparation was found to be phosphorylated on α-S487, while AMPK purified from rat liver was also phosphorylated on this site [44]. As mentioned above, ST loop phosphorylation on α-T481 downregulates α-T172 phosphorylation levels suggesting that the ST loop may act as a switch by which other signalling cascades may be able to downregulate AMPK activity. In line with this concept, α-S487 phosphorylation on a peptide corresponding to the ST loop was shown to inhibit α-LKB1-mediated T172 phosphorylation, through interaction with α1-R64, K71 or R74 to block access of LKB1 to α-T172 [47]. This is in contrast to the proposed mechanism in which α-T481 phosphorylation was thought to cause dissociation of the ST loop from the kinase domain and promote phosphatase access to α-T172 [43]. Instead of an association/dissociation mechanism, there could in fact be a more subtle conformational shift of the ST loop depending on the phosphorylation state of α-T172, α-T481 and α-S487.

Some studies have suggested that α-S487 is an autophosphorylation site, as bacterially expressed kinase active α1β1γ1 produced a α1-pS487 immunoblot signal that was abolished with substitution to Ala [66]. This is in line with previous findings that kinase dead AMPK has reduced α1-S487 phosphorylation [45] and was investigated more rigorously using both heterotrimeric AMPK and synthetic peptides corresponding to α-S487, where α1-S487 was shown to be a weak autophosphorylation site, while α2-S491 efficiently autophosphorylated [47]. In addition to autophosphorylation, multiple upstream kinases have been reported for α-S487, firstly, protein kinase B (PKB or Akt) was capable of phosphorylating purified kinase dead α1β1γ1, while insulin-induced Akt activation in perfused rat hearts and overexpression of Akt in cardiomyocytes increased α1-S487 phosphorylation [45,102]. Conversely, α2-S491 (in α2β2γ1) was later shown to be a poor Akt substrate in cell-free assay and its phosphorylation via this pathway is unlikely to be a physiologically relevant event [47]. Protein kinase A (PKA) has also been demonstrated to be an α1-S487 kinase, in cell-free assays using kinase dead α1β1γ1 [39] and in cellulo using cAMP elevating agents that stimulate PKA activity [48]. Direct phosphorylation of α2-S491 by PKA has not been reported. Additionally, phorbol esters that activate protein kinase C (PKC) were shown to stimulate α1-S487 phosphorylation in endothelial cells, and while this was shown to have no effect on α2-S491 phosphorylation levels [103], the mouse ortholog of human PKCµ, protein kinase D1 (PKD1), was shown to phosphorylate α2-S491 in C2C12 myotubes [52]. Direct phosphorylation on α1-S487 by PKC, and α2-S491 by PKD1, was confirmed under cell-free conditions [52,103]. In addition, kinases ERK1/2 [49] and IKKβ [50] have also been suggested to phosphorylate α-S487, with studies limited to in cellulo experiments using specific ERK1/2 and IKKβ inhibitors. Lastly, p70S6 kinase (p70S6K) phosphorylated α2-S491 on heterotrimeric AMPK in cell-free assays, and in cellulo in response to p70S6K-stimulating leptin. [51]. However, the specificity of ERK1/2 and IKKβ towards α1-S487 or p70S6K towards α2-S491 is unknown, as the antibody used for immunoblotting in these studies recognises both sites.

### 2.2. β-Subunit

#### 2.2.1. NH_2_-Terminal Domain

The β-subunit contains an unstructured NH_2_-terminal sequence. (NTD; β1: 1–67; β2: 1–66) that is highly divergent between β1 and β2 isoforms [104]. Within the NTD, β1-specific residues **S24** and **S25** were originally shown to be phosphorylated in AMPK purified from rat liver in a mutually exclusive manner, with β1-S24 the most abundant [53]. Phosphorylation on these sites could be enhanced by incubation of the purified rat AMPK with AMP and ATP prior to proteolytic digest, suggestive of autophosphorylation. This was later confirmed using purified heterotrimeric bacterial α1β1γ1 [62], and while PKA was also shown to phosphorylate β1-S24 under cell-free conditions [39] it is not yet known if this occurs in cells. Functionally, β1-S24/β1-S25 phosphorylation is thought to cause cytosolic distribution of the β1-subunit, as mutation of these residues to Ala dramatically increased the nuclear distribution of β1-containing AMPK complexes [54]. However, a thorough functional analysis is yet to be presented.

#### 2.2.2. CBM

Immediately following the NTD is a conserved carbohydrate binding module (CBM; β1: 68–163; β2: 67–163), that forms an interface with the N-lobe of the α kinase domain creating a hydrophobic cleft termed the allosteric drug and metabolite (ADaM) site [105]. The ADaM site binds synthetic compounds (e.g., A769662, SC4 and MK8722) [13,21,28] and endogenous long chain fatty acyl-CoAs (e.g., palmitoyl-CoA) [106], which are able to allosterically activate AMPK complexes. While MK8722 activates all 12 AMPK complexes [21], A769662 and palmitoyl-CoA are β1-selective [106,107]. Interestingly, residing in the ADaM site is **β1/2-S108**, with β1-S108 identified in AMPK purified from rat liver as an autophosphorylation site. β2-S108 may also be an autophosphorylated residue [53,57,62]. β1-S108 autophosphorylation was reported to occur in *cis* [56], whereas the autophagy initiator ULK1 was capable of phosphorylating β1-S108 but not β2-S108 [57]. Additional upstream kinases were identified in an HTP screen using a peptide corresponding to the β1-S108 motif, such as BRISK1/2, NEK2 and MLK1, however these are yet to be validated using heterotrimeric AMPK. The crystal structure of α2β1γ1 complexed to activator 991 revealed the phosphate of β1-pS108 forms electrostatic interactions with α2-K31 and possibly α2-T21, thereby stabilising the ADaM site. Thus, many AMPK drugs (e.g., A769662) have an absolute requirement for β-S108 phosphorylation to induce allosteric stimulation [55,56], whereas recent, more potent compounds (e.g., SC4) are still capable of activating β-S108A mutants, albeit at a reduced EC50 [28]. Additionally, the β1-S108A mutant has reduced basal AMPK activity and AMP Ka, and an increased AMP allosteric activation, suggesting that β-S108 phosphorylation strengthens the CBM-α kinase domain interaction and improves catalytic activity, even in the absence of ADaM site ligands [54].

The more traditional function of the CBM is to target AMPK to glycogen, an interaction we recently showed is critical for glucose handling and maximal exercise capacity [108]. The observed phenotype in this study was accompanied by a loss of total AMPK protein levels (~50–60% reduction), suggesting that the AMPK-glycogen interaction is important to stabilise certain pools of AMPK. Interestingly, a recent study utilised a tamoxifen-inducible muscle specific α1/α2 double knockout mouse model and reported impaired maximal running capacity but no effect on whole body glucose handling despite an approximately 70% reduction in α1-subunit protein levels and almost complete loss in α2-subunit protein levels [20]. Taken together, this suggests that the impaired glucose handling demonstrated in this AMPK-glycogen interaction study is likely a direct result of the compromised AMPK-glycogen binding and not due to a loss of total AMPK protein levels, while a reduction in maximal exercise capacity is likely a result of a loss of total AMPK. Intriguingly, **β-T148** lies within the carbohydrate binding pocket of the CBM, which was shown to be autophosphorylated on both α1β1γ1 and α1β2γ1 purified heterotrimer complexes, as well as in HEK293T and HepG2 cells [58]. This was confirmed using a phospho-specific antibody and mutational analysis. Phosphorylation of β-T148 was shown to disrupt the AMPK-glycogen interaction, in both cell-free glycogen immunoprecipitation experiments and in cellulo glycogen co-localisation studies, providing evidence for phosphorylation-mediated regulation of glycogen-bound pools of AMPK.

#### 2.2.3. β-Linker

Located at the extreme COOH-terminus of the β-subunit is a highly conserved α-γ-subunit binding sequence (α-γ-SBS; β1: 186–270; β2: 188–272) that forms a three-stranded antiparallel β-sheet, where the last β-strand forms an inter-subunit β-sheet with a β-strand from the γ-subunit. The α-γ-SBS is absolutely required for AMPK complex formation [109]. Located between the CBM and the α-γ-SBS is a region termed the β-linker (β1: 164–185; β2: 164–187), which includes an α-helix that interacts with the αC helix of the kinase domain, termed the αC-interacting helix (β1/2: 160–175) [13], followed by a flexible loop that leads into the α-γ-SBS. Located on this loop is **β1-S182/β2-S184** (hereafter β-S182), where β1-S182 was originally shown to be stoichiometrically phosphorylated in AMPK purified from rat liver [53], while β2-S184 was substoichiometrically phosphorylated in skeletal muscle AMPK complexes that predominantly contain the α2-subunit [59]. Early functional analysis of β-S182 phosphorylation suggests that it has a similar effect to that of β1-S24/β1-S25, increasing the cytosolic distribution of AMPK, as a β1-S182A mutation led to an increase in nuclear distribution in HEK293T cells [54]. However, the isoform specific regulation of β-S182 phosphorylation has not been investigated, and the kinase responsible for catalysing this modification is yet to be elucidated.

### 2.3. γ-Subunit

Up until recently, phospho-regulation of the regulatory γ-subunit has been largely ignored, as the majority of studies have investigated AMPK PTM regulation through the α- and β-subunits. However, a new study has recently identified two phosphorylation sites on the γ1-subunit, **γ1-S192** which is located on the linker between CBS2 and CBS3, and **γ1-T284** which is located in CBS4 [60]. These sites, although separated on the linear γ1 sequence, are in structural proximity in a solvent-exposed location. Both are conserved in the γ3-subunit (γ3-S347 and γ3-S439, respectively) but are non-phosphorylatable residues in γ2. The γ1-subunit sites were phosphorylated by purified DNA-dependent protein kinase (DNA-PK) using short peptides encompassing the γ1-S192 or γ1-T284 sites, while mutation to Ala reduced a ^32^P-incorporated autoradiogram signal in the isolated γ1-subunit. These phosphorylation events reportedly work in concert to promote lysosomal localisation, which ultimately increases LKB1-mediated α-T172 phosphorylation, without having any effect on AMP stimulation. However, further validation of direct phosphorylation of the γ1-subunit is required using heterotrimeric AMPK, while it is unknown if the γ3-subunit is phosphorylated at the equivalent sites.

## 3. Other AMPK Modifications (Non-Phosphorylation)

Refer to Table 3 for a summary of reversible, non-phosphorylation PTMs on AMPK.

### 3.1. Ubiquitination

Ubiquitination is a reversible, highly regulated PTM central to cellular signalling [137]. The process involves covalent attachment and removal (de-ubiquitination) of the 8.6 kDa protein ubiquitin (Ub) to the amino group of Lys sidechains in target proteins. The ubiquitination machinery consists of a coordinated, ATP-driven cascade involving E1 Ub-activating and E2 Ub-conjugating enzymes, as well as E3 Ub ligases. The attached Ub molecule can also be ubiquitinated on any of its seven Lys residues resulting in the formation of polyUb-chains, with various consequences for the target protein, the most common of which is targeting to the proteosome for degradation (ubiquitin/proteosome system (UPS); canonical K48 linked). Despite this, polyubiquitination of AMPK (on the α1 isoform) and AMPK-related kinases was first demonstrated as occurring via K29/K33-linkages, possibly attenuating T172 phosphorylation by LKB1 and subsequent activation [138]. Ubiquitination by a number of E3 ubiquitin ligase complexes has since been reported on AMPK isoforms α2, β1, β2 and γ1 to regulate AMPK expression and/or activity in a wide variety of settings, and on the *S. cerevisiae* AMPK ortholog Snf1 to destabilise the yeast complex and down-regulate signalling [139]. **AMPKα**-subunit ubiquitination by makorin ring finger protein 1 (MKRN1) suppresses AMPK signalling through protein destablisation, leading to reduced glucose consumption and increased lipid accumulation [110]. Consequently, MKRN1-null mice exhibit AMPK hyperactivity in liver and adipose tissue and are protected against diet-induced metabolic syndrome, whereas lowering MKRN1 expression in obese mice reversed non-alcoholic fatty liver disease. The glucose-induced degradation deficient (GID)-protein complex has been reported to directly polyubiquitinate α-subunits via K48 linkages, to induce both AMPK proteasomal degradation and suppress activity as protective mechanisms to down-regulate autophagy during periods of prolonged nutrient starvation [113]. Intriguingly, α-T172 phosphorylated AMPK appeared to be the preferred substrate for GID ubiquitination, suggesting significant conformational rearrangement upon activation to enhance exposure of the target α-Lys residue. Consistent with an AMPK suppressive role, knockdown of *gid* genes extended maximum lifespan of *C. elegans* by up to 35%. The E3 ligase cullin-RING 4A (CRL4A) polyubiquitinated, and decreased the stability of, AMPKα through non-canonical K29 and K63 linkages [112]. These modifications were enhanced in the presence of the E3 Ub ligase complex substrate receptor cereblon (CRBN) and were associated with increased secretion of allergic mediators from mouse bone marrow–derived mast cells (BMMCs). CRBN was previously demonstrated to interact with AMPKα to suppress α-T172 phosphorylation in HEK293FT cells, and mediate AMPK signalling in mouse liver in vivo to negatively influence high fat diet-induced changes to lipid and glucose homeostasis and insulin sensitivity [125,126,140]. Complimentary to ubiquitination of the α-subunit, the CRL4A-CRBN complex was subsequently shown to polyubiquitinate the **γ1-subunit** to promote its proteasomal degradation [127]. CRBN also suppressed deleterious α-T172 phosphorylation in a rat model of cerebral ischemia (middle cerebral artery occlusion/reperfusion) in response to thalidomide, a known CRBN modulator, and attenuated cardioprotective AMPK activation in mouse hearts in response to ischemia/reperfusion [141,142]. CRBN levels were found to be reduced, and AMPK levels increased, following exercise of a mouse model of type 1 diabetes, supporting a regulatory role for skeletal muscle CRBN in glucose homeostasis [143]. Finally, deficits in hippocampal-dependent learning and memory tasks in *Crbn* knockout mice associated with high AMPK activity were restored following acute pharmacological AMPK inhibition [144]. These studies highlight the potential for new therapies targeting the CRBN-AMPK interaction to treat metabolic and neurological diseases and disabilities and to provide cardio- and neuroprotection after ischemia/reperfusion or stroke.

The α-subunit has been reported to be degraded by UPS pathways associated with a PEDF/PEDFR/peroxisome proliferator activated receptor γ (PPARγ) axis, although the AMPK modifying ligase was not identified. PPARγ or proteasomal inhibition disrupted an AMPKα/PPARγ interaction, abolishing α-subunit degradation in cardiomyocytes [114]. Loss of the deubiquitinase USP10 in mouse liver in vivo and HTC116 cells has been linked to AMPK inactivation in the absence of reduced protein levels, concomitant with accumulation of lipid droplets, hepatic triglycerides and cholesterol, and increased blood glucose [111]. USP10 was found to be upregulated by glucose starvation and removed K63-linked polyubiquitin chains on **α2-K60** and **K391**, and analogous **α1-K62** and **K388**, to increase association with LKB1. Other candidate substrates are not conserved (α2-K379/K470 and α1-K276/K476), hinting at isoform-specific regulation at the deubiquitination step. In this thorough study, AMPK was also shown to activate USP10 via phosphorylation of S76, indicating the presence of a positive feedforward loop important for maintenance of metabolic homeostasis in liver.

Specific ubiquitination the **α1-subunit** by the TRIM27 ligase, directed by melanoma antigens (MAGE) A3 and A6 upregulated in many cancer states, has been demonstrated in HeLa cells and leads to AMPK degradation, reduced autophagic flux and hypersensitisation to metformin [116]. RING finger 44 (RNF44) activity also correlates with ubiquitination and proteasomal degradation of AMPKα1 in BRAFi-resistant melanoma (BR) cells, down-regulating glucose metabolism and autophagic flux following Arg starvation [117]. Besides these studies, α1 ubiquitination and UPS degradation was increased in hepatocellular carcinoma (HCC) HepG2 cells in response to intracellular PEDF, leading to elevation and suppression of lipogenic and lipolytic pathways, respectively [118]. While increases in fibroblasts and lymphocytes is observed from patients with the severe cerebral form of X-linked adrenoleukodystrophy (ALD), possibly contributing to the low bioenergetic profile of these cells [115].

AMPK α2-subunit protein levels in C2C12 myotubes are reported to be regulated by UPS pathways. α2 ubiquitination and degradation were decreased following treatment with serum obtained from calorie-restricted mice, an effect possibly controlled by the ubiquitin-specific protease USP9X that was previously shown to interact with the AMPK-related kinase NUAK1 [119,138]. Conversely, high glucose promoted UPS degradation of the α2-subunit via the E3 ubiquitin ligase, WW domain-containing protein 1 (WWP1) [120]. Vila and colleagues have presented two studies showing UPS degradation of the α2-subunit by the E2/E3 hybrid ubiquitin protein ligase UBE2O, occurring via ubiquitination of K470 [121,145]. In the first study, the UBE2O-α2 axis was postulated as a target for cancer therapies since UBE2O inhibition or *Ube2o* gene deletion restored α2-subunit levels, or impaired tumor initiation, growth and metastasis, and switched off tumor cell metabolic reprogramming in mice. The second study defined UBE2O as an essential regulator of α2-dependent glucose and lipid metabolism programs in mouse skeletal muscle, extending the therapeutic potential of the axis to metabolic disorders.

Stability of the **β1-subunit** isoform has been shown to be mediated through polyubiquitination by cell death-inducing DFF45-like effector A (CIDEA) [122]. AMPK levels, and consequently rates of fatty acid oxidation, were elevated in brown adipose tissue and differentiated adipocytes from *Cidea*^−/−^ mice, previously shown to be resistant to high fat diet-induced obesity and diabetes through increased energy expenditure [146]. Similarly, higher levels of liver β1, reduced liver lipid accumulation and partially improved liver function were reported in mice harbouring liver-specific knockout of the transcription factor E4BP40 [123]. E4BP40 was found to upregulate expression of the AMPK-specific E3 ligases RNF44, UBE2O and CRBN, potentially providing a UPS-mediated mechanism to explain the effect. In contrast, K63-linked polyubiquitination of the **β2-subunit** by a complex formed between laforin, a dual-specificity phosphatase, and malin, an E3 ubiquitin ligase, resulted in steady-state accumulation of β2, possibly due to subcellular re-localisation [124].

Ubiquitination has also been detected by multiple (>3) HTPs at α1-K387/α2-K391 and K476, and γ1-K234, K264 and K329, although the functional consequences of these PTMs are unclear [29].

### 3.2. Sumoylation

Sumoylation has many similarities to ubiquitination, except in this case Small Ubiquitin-like Modifier (SUMO, ~12 kDa) proteins are covalently attached to Lys residues on target proteins. Sumoylation does not usually target proteins for proteasomal degradation, but rather may compete for ubiquitination sites or act as a tag for subsequent UPS degradation. This reversible PTM also affects cellular processes such as apoptosis, protein stability, localisation and cell cycle progression. First evidence pointing to sumoylation of AMPK revealed the **β2** isoform, but not β1, as a substrate for the E3-SUMO ligase protein inhibitor of activated STAT PIAS4 (alternate name PIASγ) in HEK293 cells [129]. Attachment of SUMO2 units to β2 was associated with activation of a hypersumoylable AMPK mutant α2β2(K262R)γ1. Interestingly, prior ubiquitination reduced the extent of sumoylation, whereas sumoylation was antagonistic for subsequent ubiquitination and degradation by CIDEA, indicating possible competition between the two systems for substrate lysines. Soon after, Simpson-Lavy and colleagues presented evidence for glucose-induced sumoylation of Snf1 by E3-SUMO ligase Mms21 (yeast homolog of human NSE2), leading to Snf1 degradation [147]. The modified site was identified as K549 which is structurally conserved in α1 (K439) and α2 (K443), although the mammalian sites do not fit the general sumoylation motif ΦKxD/E (where Φ is a hydrophobic residue and x is any amino acid). The authors concluded that sumoylation targets Snf1 for UPS degradation by the SUMO-directed ubiquitin ligase Slx5-Slx8, and inhibits enzymatic activity by interacting with a region in close proximity to the Snf1 active site. Both **α1** and **α2-subunits** have both been shown to be sumoylated by PIAS4 in MEFs and HEK293 cells [128]. Intriguingly, PIAS4 knockdown targeted α1, but not α2, signalling towards inhibition of mTORC1 pathways via phosphorylation of TSC2. The responsible modified residue was mapped to K109 (Met in α2), located in the kinase domain E-helix. This indicates α1-sumoylation regulates AMPK substrate specificity, possibly protecting mTORC1 signalling during periods of AMPK activation. Consistent with this, PIAS4 knockdown in MDA-MD-231 breast cancer cells potentiated AMPK activity toward mTORC1 signalling and inhibited cell proliferation in response to the AMPK allosteric activator A769662.

### 3.3. Acetylation

Lysine acetylation is a reversible PTM involved in the regulation of a diverse array of cellular processes, of which the most intensely studied is gene expression through modification of core histones. Evidence for acetylation of mammalian AMPK is limited to an early study from Mitchelhill et al. [53], demonstrating acetylation of the **γ1** NH_2_-terminus in AMPK isolated from rat liver, and a handful of proteomic studies identifying acetylation of α1-K31, K33, K71/α2-K69, γ1-K264 and γ2-K9/K10. The α1 residues K31 and K33 (conserved in α2) are of particular interest as structurally they stabilise the AMPK ADaM site through interactions with the negatively charged phosphate group of phosphorylated β1-S108, shown to mediate sensitivity to A769662 and lauroyl-/myristoyl-CoAs [55,56,106]. Thus, it is tempting to speculate that charge neutralisation of these lysines by acetylation might diminish AMPK activation by long-chain fatty acid-CoA esters. Using mass spectrometry, Lu et al. identified acetylation of K12, K16, K17 and K256 in the *S. cerevisiae* AMPKβ homolog Sip2, regulated by acetyltransferase NuA4/Esa1 and deacetylase Rpd3 [148]. Sip2 acetylation diminished with age, and strains harbouring multiple acetylation mimetic mutations (Lys-to-Gln) at these positions exhibited extended lifespan and were more resistant to oxidative stress. Functionally, acetylation of the three NH_2_-terminal sites was sufficient to enhance an interaction between Sip2 and Snf1 (AMPKα) to decrease Snf1 activity. With no clear phylogenetic conservation, it remains unclear the extent to which NH_2_-terminal acetylation regulates mammalian AMPK, however it is perhaps worth noting that Sip2-K256 structurally aligns to β2-K167, found in the αC-interacting helix known to be important for AMPK activation by ADaM agonists but not AMP [13].

### 3.4. Methylation

DNA methylation is an adaptive mechanism by which skeletal muscle and other tissues respond to repeated bouts of exercise associated with endurance training. These modifications lead to altered gene transcription profiles to e.g., increased oxidative capacity and substrate utilisation, and hypertrophy, that intriguingly may be inheritable to confer beneficial effects on offspring health [149,150]. Protein methylation, the process of attaching methyl groups most commonly on nitrogens of Arg and Lys side-chains, directs a wide range of essential cellular processes but has been far less extensively studied in the context of acute or chronic exercise. Furthermore, information regarding methylation of AMPK is limited to a handful of HTP proteomics studies demonstrating methyl modifications of α2-R227, γ1-K34 and γ2-K62 [29,131,132], however the consequences of these PTMs are unknown and difficult to predict given their locations away from important AMPK regulatory elements.

### 3.5. Oxidative Modifications

Reactive oxygen species (ROS) are generated in exercising skeletal muscle via a number of processes, most notably the mitochondrial electron transport chain coupled to a rise in oxygen consumption. Exercise-generated ROS regulate many protein classes, including metabolic signalling kinases, via a suite of oxidative modifications. We direct you to a recent and extensive review concerning dynamic regulation of AMPK following generation of ROS under glucose stress [151]. In essence, oxidative AMPK PTMs are considered to be either inhibiting or activating, depending on the context, model and method of ROS generation. For example, 30 min exposure to 300 μM H_2_O_2_ stimulated AMPK in HEK293 cells but inhibited in neonatal primary cardiomyocytes [136], which the authors speculated was likely due to elevated levels of antioxidants in transformed/cancer cell lines, compared to primary cells. Oxidation of conserved residues **α2-C130** and **α2-C174** following treatment of COS7 cells with H_2_O_2_ has been reported to induce formation of an intermolecular disulphide bond, preventing association with AMPK activating kinases and suppressing signalling. The modification was negatively regulated by the oxidoreductase Trx1 (itself positively regulated by AMPK) to promote AMPK activity in the ischemic mouse heart [136]. H_2_O_2_ also reversibly inhibited activities of purified AMPK α1β1γ1 and AMPK-related kinases SIK1-3 [152]. The majority of studies, however, have demonstrated AMPK activation in response to oxidative agents or hypoxia [72,153], although the debate continues as to whether this is due to direct modification, or imbalances in AMP/ADP:ATP ratios arising from impaired mitochondrial function. Zmijewski et al. reported H_2_O_2_ directly activated the isolated AMPK kinase domain, and glucose oxidase-generated H_2_O_2_ oxidation triggered an increase in α-T172 and AMPK activation in HEK293 cells without any detectable change in ADP:ATP ratio [135]. This process possibly involved s-glutathionylation of **α1-C304** and, to a lesser extent, **α1-C299** in the AID, although regulatory roles for these residues were not detected in a subsequent report [154]. Other studies have demonstrated similar nucleotide-independent AMPK activation in response to H_2_O_2_ or hypoxia/ischemia via indirect pathways. Incubation at 1.5% O_2_ up-regulated activities of LKB1 in MEFs [155], and CaMKK2 in 143B osteosarcoma cells associated with Ca^2+^ influx and relocation of stromal interaction molecule 1- (STIM1) to the plasma membrane [156]. Intriguingly, STIM1 was identified as an AMPK substrate in L6 myotubes and validated in AMPK β1/β2 knockout MEFs, with two AMPK sites phosphorylated in both contracted rat and exercised mouse skeletal muscle [10]. Phosphorylation at STIM1-S257 was associated with altered STIM1 conformation and decreased store-operated Ca^2+^ entry, hinting at negative feedback regulating AMPK signalling during exercise. Finally, AMPK was reported to be activated independently of changes to adenine nucleotide in mild ischemic rat hearts (39% reduced coronary flow for 30 min) [157], in H_2_O_2_-incubated rat epitrochlearis muscle to induce uptake of 3-*O*-methyl-*D*-glucose [158], and in HUVEC cells incubated at low O_2_ (1.5–3%) [159], although in each case the activating kinase was not identified. It must be noted these studies reported a wide range of resting and stressed AMP:ATP ratios, possibly highlighting the difficulties associated with using HPLC to measure cellular nucleotides, for which more sensitive liquid chromatography with mass spectrometry methods have been developed [71,104].

On the other hand, several studies have attributed redox stress-induced AMPK activation to fluctuations in intracellular adenine nucleotide pools. Evans et al. reported in vivo activation of α1 AMPK in rat pulmonary arterial smooth muscle following 1 h of hypoxia was accompanied by a 2-fold increase in AMP:ATP ratio [160]. Exogenously applied, or glucose oxidase generated, H_2_O_2_ also induced elevated AMP:ATP ratios and AMPK signalling in cultured NIH3T3 or HEK293 cells, respectively [161,162]. Accordingly, H_2_O_2_-induced AMPK signalling was blunted in HEK293 cells stably expressing an AMP-insensitive γ1-R531G mutant [162,163]. More comprehensive studies are clearly needed to delineate the AMPK response to oxidative stress in a variety of physiological scenarios.

**α1-C176** and **α1-C416** were found to be s-nitrosylated in treated HEK or human pulmonary arterial endothelial cell (HPAEC) lysates, and a range of Cys residues in all AMPK isoforms, with the exception of γ3, were modified in mouse tissues and human cell lines, although functional consequences remain unknown [133,134,164].

AMPK is one of 102 human Ser/Thr kinases with a Cys residue at position +2 in relation to the activating phosphorylation site in the kinase domain activation loop [152]. These kinases represent ~11.5% of the human kinome and are almost exclusively limited to the CAMK and AGC groups. Oxidative modification of Cys^+2^ was found to negatively regulate activity of human Aurora A (C290), supporting previous in vitro and in vivo studies examining PKA (C199) [165]. Using a bacterial expression system, we previously found that mutation of the AMPKα1-Cys^+2^ (α1-C176) to Ser prevented accumulation of autophosphorylated species following induction, without deleteriously affecting allosteric regulation [57]. One explanation for this phenomenon is that oxidative modification of α1-C176 imparted recombinant complexes with low intrinsic activity, manifesting as autophosphorylation in the bacterial environment devoid of phosphatase pressure. Further work is clearly needed to determine the regulatory roles, if any, of AMPKα-Cys^+2^.

## 4. Perspectives

AMPK is an ancient energy sensor that calibrates cellular growth potential with nutrient availability. Given its standing in the governance of cellular metabolism, it is hardly surprising that AMPK succumbs to a multiplicity of PTMs whose diversity (i.e., phosphorylation, oxidation, ubiquitination etc.) is likely reconciled by the heterogeneity of AMPK expression profiles in distinct cells and tissues.

As discussed, the overwhelming majority of investigations of posttranslational regulation of AMPK to date have investigated α- and β-subunits, with independent studies of the γ-subunit dominated by its conventional nucleotide-sensing capacity. However, if we wish to fully understand how signalling events like phosphorylation cascades control AMPK throughout the body, and the implications in for health and disease, then it would be remiss to ignore that both γ2 and γ3 isoforms are characterised by unique, 241 and 168 amino acid NH_2_-terminal extensions, respectively. A total of 46 putative phospho-sites, conserved between mouse and human species, have been flagged in these in these two γ2 and γ3 regions in HTP studies. Notwithstanding that species conservation of only 2 sites exist in γ3, this region in humans still harbours a total of 34 Ser/Thr residues, suggesting some have evolved to cater for the differential metabolic requirements of human skeletal muscle [166], the sole tissue housing γ3. Given γ3-complexed AMPK is generally insensitive to allosteric stimulation by adenine nucleotides [167], it stands to reason that posttranslational phosphorylation of its NH_2_-terminal extension is one such mechanism modulating activity. It is also noteworthy that of the two longer γ isoforms, γ2 contains 14 Ser-Pro sites, which is significant when one considers the critical role of proline-directed kinases in cell cycle regulation and their intense interest for the cancer and neurodegeneration fields [168,169]. In fact, the γ2 NH_2_-terminal extension contains nuclear localisation and export sequences, whereby γ2-containing complexes were found to phosphorylate and inhibit the nuclear transcription factor TIF-IA to prevent ribosome biogenesis [170], a process antagonistically controlled by mTORC1 [171]. This alludes to γ2-expressing AMPK occupying a position of prominence in cell cycle regulation and opens up the possibility of its phosphorylation controlling nuclear cycling and/or activity. With the exception of phosphorylation of the β-S182 residue, also a Pro-directed site that regulates nuclear localisation of AMPK [54], these seemingly vital roles have otherwise been overlooked in the context of AMPK research; γ2 is ubiquitously expressed and the hitherto vast majority of studies on this isoform have been confined to a single organ, the heart.

Finally, if the field as a whole seeks to move forward in its collective understanding of AMPK regulation, greater rigour in regulatory site identification needs to be respected. For example, full-length heterotrimeric AMPK as a substrate, as opposed to protein fragments, are a must for phosphorylation reactions under cell-free conditions, preferably over a shorter duration more indicative of the rapidity of cellular signalling. Promiscuous kinases may hit multiple sites on purified enzyme substrates, with little-to-no applicability to complex biological systems in cells. Taken into account, we need more orthogonal approaches to PTM validation involving a variety of techniques, such as mass spectrometry, stringent phospho-antibody validation, and confirmation in heterogenous cell lines. Furthermore, conclusions should not be solely based on overexpression and/or knockdown effects in cells; while many inhibitors have off-target effects, and these need to be considered when analysing changes in site regulation. Without doubt this creates challenges for researchers in the field but will ultimately be advantageous to provide greater clarity on the regulation of AMPK biology.

## Figures and Tables

**Figure 1 ijms-22-01229-f001:**
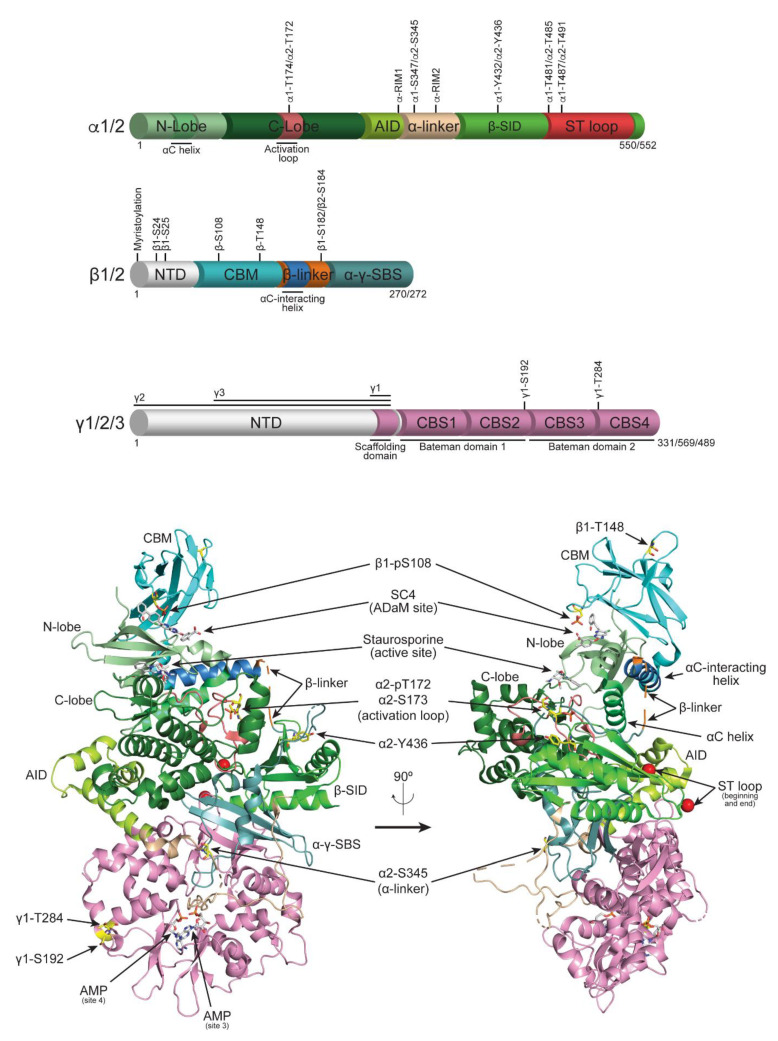
**Schematic representation of subunit domain organisation and cartoon representation of the AMPK quaternary structure**. The AMPK heterotrimer consists of a catalytic-regulatory α-subunit (green), and regulatory β- (blue) and γ- (magenta) subunits. The main structural features are shown above as a linear tube representation, with indication to the location of phosphorylation sites with known function. The varying lengths of the γ-subunit NTD’s are depicted above the γ-subunit domain schematic. The crystal structure of the α2β1γ1 in complex with SC4 (PDB: 6B1U) is shown below as a cartoon representation and the subunit features are colour coded to that of the tube schematics above. SC4 (white sticks) is shown bound to the ADaM-site created at the interface of the CBM and the N-lobe of the kinase domain. Staurosporine (white sticks) is shown bound to the active site between the N- and C-lobes of the kinase domain. Two AMP molecules (white sticks) are shown bound to the γ-subunit in sites 3 and 4. Phosphorylated α2-T172 (α2-pT172) and β1-S108 (β1-pS108), and non-phosphorylated α2-S173, α2-S345, α2-Y436, β1-T148, γ1-S192 and γ1-T284 are shown as yellow sticks. The flexible ST loop, β-subunit NTD and β-linker regions that all contain functional phosphorylation sites are not modelled in any current AMPK crystal structures. The beginning and end of the ST loop is shown here as red spheres. The image shown on the right is a 90° rotation (to the left) of the image on the left; image generated using PyMol.

**Table 1 ijms-22-01229-t001:** AMPK phosphosites: regulatory function assigned.

Isoform/Residue	Location	Regulatory Function	Notes	Kinase	Primary Ref.
**α-subunit** Uniprot IDs α1: Q13131-1 (numbering starts from initiation Met10, most commonly used in the field); α2: P54646-1					
α1-T174/α2-T172 [30]	Activation loop	Activatory	Fully validated upstream kinases	LKB1	[31,32]
				CaMKK2	[33] [34]
			Less well validated upstream kinases	TAK1	[35,36]
				MLK3	[37]
				CaMKK1	[34]
				VRK1	[38]
α1-S175/α2-S173	Activation loop	Inhibitory	Suppresses T172 phosphorylation	PKA	[39]
α1-S347/α2-S345	α-linker	Inhibitory	Suppresses T172 phosphorylation	CDK4	[40]
				mTORC1	[41]
α1-Y432/α2-Y436	β-SID	Inhibitory		Fyn	[42]
α1-T481/α2-T485	ST loop	Inhibitory	Promotes T172 dephosphorylation	GSK3α/β	[43]
				CDK4 (only α2-T485?)	[40]
α1-S487/α2-S491 [44]	ST loop	Inhibitory	Suppresses T172 phosphorylation	autophos. (α2-S491 more than α1-S487)	[45]
				Akt/PKB (α1-S487 more than α2-S491)	[46] [45] [47]
				PKA (only α1-S487)	[39,48]
				PKC (only α1-S487?)	[44]
				ERK1/2	[49]
				IKKβ	[50]
				p70S6K	[51]
				PKD1 (only α2-S491?)	[52]
**β-subunit** Uniprot IDs β1: Q9Y478-1; β2: O43741-1					
β1-S24/25 [53]	NH_2_-terminus	Localisation	S24/25A promotes nuclear transport [54]	Akt/PKB	[39]
β1/2-S108 [53]	CBM	Activatory	Stabilises ADaM site and sensitises AMPK to ligands [13,55,56]; β1-S108A reduced basal activity [54]	*cis*-autophos.	[53]
				ULK1 (β1-S108 only)	[57]
β1/2-T148	CBM	Localisation	T148D destabilises glycogen complex	autophos.	[58]
β1-S182/ β2-S184 [53]	β-linker	Localisation	β1-S182A promotes nuclear transport; skeletal muscle: β1-pS182 stoichiometric; β2-pS184 substoichiometric, 41 HTPs		[53,54,59]
**γ-subunit** Uniprot ID γ1: P54619-1					
γ1-S192	CBS2-3 linker	Activatory	γ1-S192A (with γ1-T284A) attenuates lysosome transport	DNA-PK	[60]
γ1-T284	CBS4	Activatory	γ1-T284A (with γ1-S192A) attenuates lysosome transport	DNA-PK	[60]

**Table 2 ijms-22-01229-t002:** AMPK phosphosites: regulatory function not assigned.

Isoform/Residue	Location	Notes	Kinase	Primary Ref.
**α-subunit** Uniprot IDs α1: Q13131-1 (numbering starts from initiation Met10, most commonly used in the field); α2: P54646-1				
α1-S178/α2-S176	Activation loop	18 HTPs		[29]
α1-T260	Kinase domain C-lobe	In vitro phosphorylation	autophos.	[44]
α1-S351/T359	α-linker	In vitro phosphorylation	ULK1	[61]
α1-T373	α-linker	30 HTPs		[29]
α2-S377	α-linker	28 HTPs; in vitro phosphorylation	CDK4	[29,40]
α1-T379	α-linker	In vitro phosphorylation	autophos.	[62]
α1-S388	α-linker	In vitro phosphorylation	ULK1	[61]
α1-S473	ST loop	In vitro phosphorylation	GSK3α/β	[43]
α1-S477	ST loop	In vitro phosphorylation	GSK3α/β	[43]
			ULK1	[61]
α1-T479	ST loop	In vitro phosphorylation	ULK1	[61]
α1-S497	ST loop	In vitro phosphorylation	PKA	[39]
α1-S499	ST loop	27 HTPs ; in vitro phosphorylation	PKA	[29,39]
α2-S529	ST loop	In vitro phosphorylation	CDK4	[40]
**β-subunit** Uniprot IDs β1: Q9Y478-1; β2: O43741-1				
β2-S39	NH_2_-terminus	In vitro phosphorylation; 22 HTPs	ULK1	[61]
β2-T40	NH_2_-terminus	In vitro phosphorylation	ULK1	[61]
β1-S49	NH_2_-terminus	In vitro phosphorylation	ULK1	[57]
β2-S69	CBM	In vitro phosphorylation	ULK1	[61]
β1-T80	CBM	In vitro phosphorylation	autophos.	[62]
β1-S96	CBM	In vitro phosphorylation	autophos.	[44]
			ULK1	[57]
β1-S101	CBM	In vitro phosphorylation	autophos.	[44]
β1-T158	CBM	In vitro phosphorylation	autophos.	[62]
β1-S170	αC-interacting helix	In vitro phosphorylation	ULK1	[57]
β1/2-S174	αC-interacting helix	In vitro phosphorylation	autophos. (β1-S174)	[62]
			ULK1 (β2-S174)	[61]
β1-S177	β-linker	In vitro phosphorylation	autophos.	[62]
**γ-subunit** Uniprot IDs γ1: P54619-1; γ2: Q9UGJ0-1				
γ1-S261/T263	CBS3	In vitro phosphorylation	ULK1	[57,61]
γ1-S270	CBS3-4 linker	In vitro phosphorylation	ULK1	[57,61]
γ1-Y272	CBS3-4 linker	65 HTPs		[29]
γ2-S65	NH_2_-terminus	33 HTPs		[29]
γ2-S71	NH_2_-terminus	25 HTPs		[29]
γ2-S122	NH_2_-terminus	16 HTPs		[29]
γ2-S196	NH_2_-terminus	25 HTPs		[29]

**Table 3 ijms-22-01229-t003:** Reversible AMPK modifications (non-phosphorylation).

Isoform/Residue	Location	Notes	Modifying Enzyme	Primary Ref.
**Ubiquitination**				
α1/α2		Systemic metabolism	MKRN1	[110]
α1-K62/276/388/476 α2-K60/379/391/470		K63-linked deubiquitination promotes AMPK activity	USP10	[111]
α1/α2		Increased allergic mediator secretion from BMMCs	CRL4A, enhanced by CRBN	[112]
α1/α2		Suppression of autophagy during prolonged starvation	GID ubiquitin ligase complex	[113]
α1/α2		Mediated by PEDF/PEDFR/PPARγ axis		[114]
α1		Reduced AMPK expression and bioenergetic profile in ALD fibroblasts and lymphocytes		[115]
α1		Cancer-specific	MAGE-A3/6-TRIM28	[116]
α1		Promotes α1 degradation in BRAFi-resistant melanoma cells	RNF44	[117]
α1		PEDF promotes α1 degradation to disturb lipid metabolism in HCC cells		[118]
α2		USP9X-mediated α2 ubiquitination in C2C12 cells suppressed by calorie restriction serum		[119]
α2		Downregulates α2 in C2C12 cells	WWP1	[120]
α1-K387/α2-K391	α-linker	4 HTPs		[29]
α1-K476	ST loop	5 HTPs		[29]
α2-K470	β-SID	Cancer progression	UBE2O	[121]
β1		Promotes β1 degradation in brown adipose tissue and adipocytes to suppress fat oxidation	Promoted by Cidea	[122]
β1		Induced by E4BP4, promoting degradation		[123]
β2		K63-linked Ub chains; accumulation, possibly associated with localisation	Laforin-malin complex	[124]
γ1		Promotes γ1 degradation through interaction with α, exacerbating high fat diet-induced metabolic defects	CRL4A^CRBN^	[125,126,127]
γ1-K234	CBS3	6 HTPs		[29]
γ1-K264	CBS3	7 HTPs		[29]
γ1-K329	CBS4	6 HTPs		[29]
**Sumoylation**				
α1-K109/α2	Kinase domain C-lobe	Down regulates AMPK activity towards mTORC1 signalling	PIAS4	[128]
β2		Increases AMPK activity; competes with ubiquitination	PIAS4	[129]
**Acetylation**				
α1-K31	Kinase domain N-lobe (ADaM site)	K31 stabilizes ADaM site		[130]
α1-K33	Kinase domain N-lobe (ADaM site)	K33 stabilizes ADaM site		[130]
α1-K71/α2-K69	Kinase domain C-helix	4 HTPs		[29]
γ1	NH_2_-terminus	Deduced from rat liver prep ToF-MS	NatB complex?	[53]
γ1-K264	CBS3	3 HTPs		[29]
γ2-K9/K10	NH_2_-terminus	1 HTP		[130]
**Methylation**				
α2-R227	Kinase domain C-lobe	1 HTPs; monomethylation		[29,131]
γ1-K34	NH_2_-terminus	1 HTP		[29,132]
γ2-K62	NH_2_-terminus	1 HTP		[29]
**Oxidation**				
α1-C176	Activation loop	s-nitrosylation; in vitro using HPAEC and HEK lysates		[133,134]
α1-C299/304	AID	Activating; C304 may be a specific target for s-glutathionylation		[135]
α1-C416	β-SID	s-nitrosylation; in vitro using HEK lysates		[134]
α2-C130/174	Kinase domain C-lobe	Inhibitory; interferes with AMPK kinase interactions. Suppressed by Trx1		[136]

## Data Availability

Not applicable.

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
