# Peer review of "Post-Translational Modifications of the Energy Guardian AMP-Activated Protein Kinase"

_ijms, 2021, doi:10.3390/ijms22031229_

Round 1

Reviewer 1 Report

In the presented manuscript “Post-translational modifications of the energy guardian AMP-activated protein kinase” written by Ovens et. al., the authors summarized the actual information about post-translation modifications of AMPK, highlighted current knowledge gaps in mentioned area and provided perspectives on how the field can apply greater rigor to the characterization of novel AMPK regulatory. The review article is well written and the flow of the paper was good. Taken all this together the manuscript deserves a minor revision from the authors to be published in an International Journal of Molecular Sciences.

Major Points

  • Mentioned abstract suggests that authors are describing not only post-translational modifications of AMPK but also their relationship to physical exercise. Therefore, authors should consider a more detailed description of the mentioned link, which would be very interesting for the readers, it would provide new perspective in the AMPK research, and it would distinguish mentioned review article from others, which are similarly focused on post-translation modifications of AMPK. Abstract in the mentioned form is misleading. Is it promising information, which is not presented in the text or they are insufficient. Some suggestions are reported below:
    • Authors are reviewing the role of complex substrate receptor cereblon (CRBN), which could interact with αAMPK resulting in suppression of α-T172 phosphorylation. Authors are reported that CRBN could influence negatively not just lipid and glucose homeostasis but also could attenuate cardio-protective effect of AMPK. Despite the fact that physical exercise due to AMPK-dependent mechanisms is playing a protective role in the cardiovascular system it would be interesting to describe the link between CRBN and exercise.
    • Similarly, several studies pointed to changes of DNA methylation caused by physical activity. Authors should describe the mentioned association exercise – methylation.
    •  
  • Authors are describing oxidative modifications of AMPK. Generally, I consider the mentioned part insufficient and authors should consider describing oxidative modifications of AMPK in more detail.
  • Authors should consider including of α1-selective small molecule activator of AMPK –Compound 13 to the text (PMID: 25036776; 32560060) and selective activator of γ1 AMPK subunit – PT-1 (PMID: 25695398; 18321858) to the text.

Minor Points

  • Please review the manuscript for typos and grammar mistakes. Some minor comments and typos (not the only ones) are reported below:

Page 7, Line 7: “substate”

Page 8, Line 42: “signalling which”  → “signaling, which”

Page 17, Line 4-5: “Combined, a total of 4 putative phospho-sites,…”  → “Combined, 4 putative phospho-sites,…”

Page 17, Line 27: “More promiscuous kinases…”  → “Kinases that are more promiscuous…”

Page 17, Line 33: “No doubt this creates …” → ““No doubt, this creates …”

  • Authors should unify the use of terms - “in cellulo” and “in vitro” for experiments using cell cultures. Moreover on page 9, line 15 are mentioned both, authors should use just one.

Reviewer 2 Report

In this review Ovens et al., provide a detailed and comprehensive description of the post-translational modifications of AMPK including phosphorylation, ubiquitination, sumoylation, acetylation, methylation and oxidation.

I agree with the authors on the merits of such a review in an exercise issue. Clearly a lot of new hypotheses may be generated from reading and thinking about other PTMs than the classic exercise-induced phospho changes.  However, I think the gain that readers may have from reading this piece could be presented more clearly.
Especially from the introduction I believe the benefits that readers interested in exercise signaling pathways may have from reading the review should appear better. The introduction in its current form also seems a bit unfocussed and lack a bit of flow, eg. the PTMs flies in at the last 7 lines.
Furthermore, although many of the PTMs described have not been studied in the context of exercise I think that it is also crucial that the authors try to associate the PTMs with exercise signaling and attempt to explain the likelihood for the different PTMs to be relevant in an exercise context.

In other words, my main points for improvements are
1) modify the introduction to improve the flow, better inform the target reader of the benefits of reading the review and to establish the relevance of your main text
2) modify the main text throughout to associate it better to exercise signaling pathways.

Minor and specifics:

Given the complexity and differential regulated AMPK signaling in different cell types and tissues it will be preferable if the authors more consistently specify the referenced cell models, eg. at p.8 l. 34 where the authors write “mammalian cells” the specific cell lines studied should be mentioned. This is just an example. I think it will improve the review if the aspect of cell line/tissue differences in signaling response is taken more carefully into consideration throughout the review.   

P. 2, l. 3-7: I think it would be nice to also reference the original Bergstöm and Hultman study identifying the concept.

P. 2, l. 21-26: It is unclear what is meant by gamma3 is exclusively expressed in white glycolytic muscles, please clarify. See for instance this study (PMID: 25640469) finding gamma3 in type I fibers, albeit in a lower expression than in type II fibers. Also worth mentioning, the requirement of gamma3 and AMPK in muscle glucose uptake during exercise as well as the role of AMPK for lipid oxidation during exercise is debated (PMID: 32504885, PMID: 31010958).

P. 2, l. 31-33: Difficult to understand what is meant by “this notion” and “sever this link”

P. 8, l. 17: Perhaps worth mentioning that SBI-0206965 has been shown to inhibit Ulk1 as well since this may affect the interpretation of the data.

P. 8, l. 29: “C2C12 myotubes and mouse muscle cells” is in terminology the same thing.

P. 8, l. 39-43: It is worth discussing the rodent AMPK transgenic data in this context that may question the causation of the human observations (PMID: 19237506, PMID: 20460585, PMID: 32372406). Also whether mTORC1 is rate limiting for contraction-mediated (you write contraction-dependent which is maybe not the most accurate wording) protein synthesis is questionable see eg. PMID: 27502839.

P. 11, l. 4-5: In regards to your recent cool study of AMPK glycogen interaction it would be informative to have your take on the reduction of total AMPK and AMPK activity vs. the interaction per se as the cause for the regulated glucose handling and running capacity.    
